# Orientation-Aware Diffusion Super-Resolution for 3T-Like Fetal MRI from Routine 1.5T Scans

**Xinliu Zhong**[1,2]                          XINLIU.ZHONG@EMORY.EDU
**Ruiying Liu**[2]                                RLIU60@EMORY.EDU
**Guohao Lin**[2]                        LINGUOHAO111@GMAIL.COM
**Chuan Huang**[3]                        CHUAN.HUANG@EMORY.EDU
**Adam Ezra Goldman-Yassen**[3,4]     ADAM.EZRA.GOLDMAN-YASSEN@EMORY.EDU
**Amy Robben Mehollin-Ray**[3,4]     AMY.ROBBEN.MEHOLLIN-RAY@EMORY.EDU
**Yun Wang**[2]                              YUN.WANG2@EMORY.EDU

[1] *Department of Computer Science, Emory University, Atlanta, GA, USA*

[2] *Department of Biomedical Informatics, Emory University, Atlanta, GA, USA*

[3] *Department of Radiology and Imaging Sciences, Emory University, Atlanta, GA, USA*

[4] *Children's Healthcare of Atlanta, Atlanta, GA, USA*

**Editors:** Accepted for publication at MIDL 2026

## Abstract

Fetal MRI plays a central role in assessing early brain development. While 3T scanners offer higher SNR and improved cortical detail, their increased sensitivity to motion, susceptibility artifacts, and $B_1$ inhomogeneity limits wide adoption for routine fetal imaging. Consequently, most clinical examinations are performed at 1.5T, where greater motion tolerance comes at the cost of lower SNR, reduced gray-white matter contrast, and partial-volume blurring - factors that undermine downstream morphometric analysis. Bridging this quality gap without sacrificing motion robustness of 1.5T would enable 3T-like morphometric reliability in routine clinical acquisitions.

We propose an orientation-aware diffusion super-resolution framework that synthesizes 3T-like fetal brain contrast from routine 1.5T scans. The model combines a Swin-UNet backbone with gated FiLM-based orientation embeddings and a residual error-shifting diffusion mechanism. Training leverages the FaBiAN phantom to generate controllable high-/low-resolution pairs with monotonic intensity remapping, geometric perturbations, and simulated signal voids, thereby ensuring generalization to clinical data. Our model produces markedly sharper gyri and mitigates partial-volume effects in both synthesized and clinical data. When evaluated using Fetal-SynthSeg following NeSVoR reconstruction, the framework consistently improves tissue segmentation accuracy over state-of-the-art restoration baselines, yielding more reliable morphometric estimates for fetal brain analysis.

**Keywords:** MRI, Diffusion Models, Image Enhancement, Fetal Neuroimaging

## 1. Introduction

MRI is widely used in neuroimaging due to its non-invasive nature, excellent soft tissue contrast, and painless procedure. Fetal brain assessment still depends largely on 1.5T scanners even though 3T acquisitions provide sharper cortical detail, higher SNR, and better gray–white separation. Consequently, routine fetal scans often suffer from an "effective

resolution" gap: although the nominal pixel size may be sufficient, the inherently lower SNR at 1.5T results in partial-volume blurring and noise that obscure fine anatomical details. This degradation complicates diagnostic tasks, such as detecting cortical dysplasia, and hinders longitudinal studies that attempt to harmonize data across varying field strengths (Jannat et al., 2025; Zimmermann, 2025). Bridging this gap via software—translating 1.5T scans to 3T quality—is therefore critical for modernizing fetal neuroimaging without costly hardware replacements.

Existing post-acquisition enhancement strategies falter under fetal-specific constraints. Classic model-based Super-Resolution (SR) methods rely on handcrafted priors (e.g., total variation, sparsity), which must precisely match the physical degradation to avoid artifact amplification. Supervised deep learning methods demand perfectly aligned low-/high-field pairs that are practically unobtainable in fetal cohorts, while physics-informed reconstructions require raw k-space data rarely archived during routine HASTE exams. While Denoising Diffusion Probabilistic Models (DDPMs) offer a generative alternative, standard implementations are prone to structural hallucinations when trained on limited medical datasets (Khateri et al., 2025). Even recent residual-shifting approaches—such as ResShift (Yue et al., 2024) or Res-SRDiff (Safari et al., 2025)—fail to account for the complex acquisition geometry, treating every slice as an independent, isotropic image.

This geometric oversight is critical. Routine fetal exams consist of orthogonal stacks (axial, coronal, sagittal) with highly anisotropic resolutions and distinct, view-dependent artifact patterns. Orientation-agnostic networks inevitably average these incompatible priors, resulting in suboptimal smoothing. To address this without clinical ground truth, we leverage high-fidelity simulation. Simulators like FaBiAN (Lajous et al., 2022) provide the only viable source of registered supervision, allowing us to explicitly learn these orientation-dependent degradations where reacquiring paired clinical data is impossible.

We address these gaps with an orientation-aware diffusion framework tailored to fetal MRI enhancement. Our contributions are threefold: (1) We introduce a gated FiLM orientation encoder that conditions the network on slice geometry, allowing it to adaptively invert view-specific anisotropies at different feature depths. (2) We propose a residual-shift diffusion formulation that anchors the generative process to the input, refining high-frequency details while explicitly mitigating the risk of hallucination common in standard DDPMs. (3) We introduce a multi-level augmentation suite—including monotonic intensity remapping, geometric perturbations, and blackout-style motion corruption——to robustly generalize from FaBiAN synthetic supervision to clinical scans. Together, these components deliver 3T-like fidelity with sharper cortical detail and improved downstream segmentation utility.

## 2. Related Work

### 2.1. Deep Learning-based Super-Resolution for Medical Imaging

Modern SR has evolved from residual CNNs to high-capacity architectures like Real-ESRGAN (Wang et al., 2018) and Vision Transformers like SwinIR (Liang et al., 2021), which utilize adversarial training or shifted-window attention to capture complex textures. Recently, state-space models such as GAMBAS (Baljer et al., 2025) have introduced Mamba layers for volumetric context, while BME-X (Sun et al., 2025) establishes a unified foundation model for multi-task restoration. However, these approaches face distinct limitations in fetal imag-

ing. Deterministic regressors (e.g., SwinIR) tend to suppress high-frequency details—the "regression to the mean" effect—particularly when pixel-aligned supervision is unavailable due to stochastic fetal motion. Furthermore, while GAMBAS assumes consistent volumetric inputs, it struggles with severe inter-slice motion of HASTE stacks. Similarly, BME-X's general-purpose design prioritizes global harmonization but lacks explicit orientation conditioning, often leading to over-smoothed results that fail to resolve the view-dependent anisotropy inherent to single-shot acquisitions.

## 2.2. Diffusion Models for Image Restoration

DDPMs (Ho et al., 2020; Saharia et al., 2023) have surpassed deterministic baselines by synthesizing the high-frequency textures essential for perceptual quality. This advantage has enabled successful applications in medical image reconstruction and harmonization (Chung and Ye, 2022; Peng et al., 2022; Özdenizci and Legenstein, 2023). However, standard DDPMs face critical hurdles in clinical deployment: generating anatomy from pure Gaussian noise is computationally intensive and prone to structural hallucinations, particularly given the domain shift between synthetic training data and real clinical scans. Furthermore, generic diffusion processes lack explicit priors to handle structured, view-dependent artifacts of fast HASTE sequences. We address these limitations by adopting a residual-shift formulation; rather than synthesizing images from scratch, our model iteratively refines high-frequency residual relative to input. This significantly constrains the generative search space, ensuring anatomical fidelity while recovering fine details.

## 2.3. Orientation as a Conditioning Signal

In fetal MRI, the trade-off between acquisition speed and spatial resolution necessitates highly anisotropic voxel dimensions, resulting in through-plane resolution (typically 3–4 mm) that is substantially coarser than the in-plane resolution ($\approx 1.0$ mm). While Slice-to-Volume Reconstruction (SVR) (Kuklisova-Murgasova et al., 2012) mitigates this by fusing orthogonal stacks, its success depends critically on the fidelity of input slices. Our work therefore targets the acquisition space *prior* to SVR: by enhancing in-plane resolution of individual stacks, we aim to stabilize subsequent registration and fusion.

From a modeling perspective, slice orientation is often treated as an implicit nuisance factor or handled only through geometric constraints. However, for texture-sparse medical images, deep networks lack sufficient visual cues to reliably infer orientation-dependent degradation kernels purely from appearance (Huang et al., 2024). To address this limitation, several explicit conditioning strategies have been explored. Conditional normalization mechanisms such as FiLM (Feature-wise Linear Modulation) (Perez et al., 2018) enable a shared backbone to adapt its feature responses to view-specific statistics, while view-specific 2D processing strategies, such as those adopted in QuickNAT (Roy et al., 2019), explicitly decouple axial, coronal, and sagittal feature distributions. In the broader diffusion literature, more heavyweight conditioning paradigms have also emerged, including parallel control-branch architectures such as ControlNet (Zhang et al., 2023) and expert-routing schemes based on Mixture-of-Experts (MoE) (Shazeer et al., 2017). While effective for strong external control or large-capacity modeling, these approaches introduce substantial architectural overhead and are not optimized for the specific constraints of fetal MRI.

In fetal HASTE acquisitions, orientation-aware conditioning is paramount: axial stacks emphasize ventricles, coronal stacks compare hemispheres, and sagittal stacks delineate midline structures, each exhibiting distinct artifact patterns and partial-volume effects. A uniform, orientation-agnostic model inevitably averages these conflicting priors, leading to degraded anatomical fidelity. In contrast, our approach treats orientation as a dynamic and explicit conditioning signal, using gated modulation to selectively activate view-specific priors at only the network depths where they are anatomically relevant.

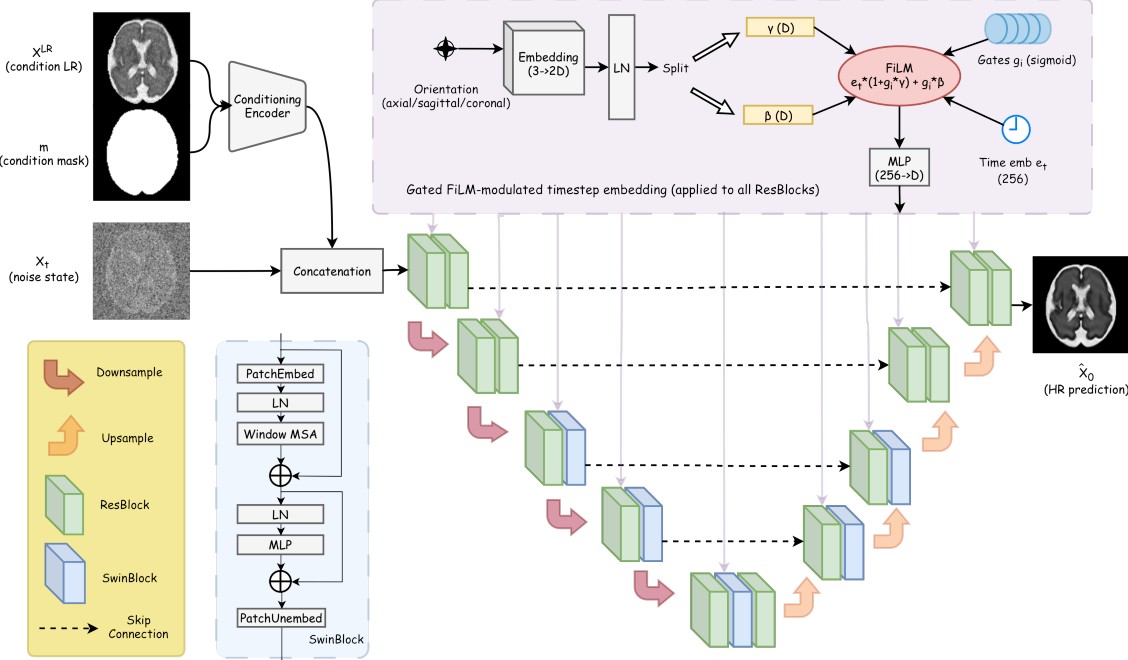

Figure 1: Overview of our framework. The architecture employs a dual-stream strategy: the Conditioning Encoder extracts spatial guidance features from the fixed low-resolution reference $\mathbf{x}^{\mathrm{LR}}$ and validity mask $\mathbf{m}$. These features are concatenated with the noisy latent state $\mathbf{x}_t$ and passed to the Swin-UNet backbone, which predicts the clean high-resolution estimate $\hat{\mathbf{x}}_0$. Gated FiLM layers inject global orientation priors, dynamically adapting the feature hierarchy to view-specific acquisition characteristics.

## 3. Methods

Our proposed framework integrates an orientation-aware hybrid Swin-UNet backbone, $f_\theta$, with a residual-shift diffusion process (Figure 1). The network processes inputs in two streams: a shallow conditioning encoder extracts the clean reference slice $\mathbf{x}^{\mathrm{LR}}$ and validity mask $\mathbf{m}$, which are concatenated with noisy diffused input $\mathbf{x}_t$ and passed through a four-stage U-Net encoder-decoder. To balance local texture recovery with global coherence, we employ standard Residual Blocks at higher resolutions and Swin Transformer Blocks at bottleneck levels, linked via skip connections. A global orientation embedding modulates features at every stage via gated FiLM layers. By operating directly on the residual mani-

fold, this architecture focuses generative capacity on restoring high-frequency details while preserving the low-frequency structure provided by input.

## 3.1. Residual-Shift Diffusion Process

We adopt the Res-SRDiff formulation (Safari et al., 2025), which extends ResShift (Yue et al., 2024) to medical image restoration. We define our inputs as follows: $\mathbf{x}^{\text{LR}}$ denotes the fixed low-resolution reference slice (the condition), $\mathbf{x}^{\text{HR}}$ represents the high-resolution target, and $\mathbf{m}$ is the binary validity mask indicating non-background regions. The diffusion process operates on the residual manifold. The forward process, detailed in Algorithm 1, perturbs the clean residual $\mathbf{r}_0 = \mathbf{x}^{\text{HR}} - \mathbf{x}^{\text{LR}}$ into a noisy latent state $\mathbf{x}_t$ at timestep $t$. The noisy observation $\mathbf{x}_t$ is defined as:

$$\mathbf{x}_t = \mathbf{x}^{\text{LR}} + \sqrt{\bar{\alpha}_t}(\mathbf{x}^{\text{HR}} - \mathbf{x}^{\text{LR}}) + \sqrt{1 - \bar{\alpha}_t}\,\boldsymbol{\epsilon}, \quad \boldsymbol{\epsilon} \sim \mathcal{N}(\mathbf{0}, \mathbf{I}). \tag{1}$$

Here, $\bar{\alpha}_t$ follows a noise schedule. At each timestep $t$, the backbone $f_\theta$ predicts the clean high-resolution estimate $\hat{\mathbf{x}}_0$. Crucially, to guide this generation, the network receives a channel-wise concatenation of the noisy state $\mathbf{x}_t$ and the conditioning features encoded from the reference pair $(\mathbf{x}^{\text{LR}}, \mathbf{m})$. We set the prediction target to the clean signal (`predict_type = xstart`), effectively training the network to recover the residual $\hat{\mathbf{r}}_0 = \hat{\mathbf{x}}_0 - \mathbf{x}^{\text{LR}}$, anchoring generation to the acquired anatomy. The reverse sampling process, presented in Algorithm 2, iteratively refines a noisy initial state $\mathbf{x}_T$ back to the clean estimate $\mathbf{x}_0$.

| **Algorithm 1:** Training | **Algorithm 2:** Sampling |
|---|---|
| **Input:** $(\mathbf{x}^{\text{LR}}, \mathbf{x}^{\text{HR}}, \mathbf{m}) \sim \mathcal{D}$ | **Input:** $\mathbf{x}^{\text{LR}}, \mathbf{m}$ |
| **repeat** | $\mathbf{x}_T \leftarrow \mathcal{N}(\mathbf{x}^{\text{LR}}, \gamma^2 \beta_T \mathbf{I})$; |
| $\quad$ Draw $t \sim \mathcal{U}(1, T)$, $\boldsymbol{\epsilon} \sim \mathcal{N}(\mathbf{0}, \mathbf{I})$; | **for** $t \leftarrow T; t \geq 1; t \leftarrow t-1$ **do** |
| $\quad \mathbf{x}_t \leftarrow$ | $\quad \epsilon \leftarrow \begin{cases} \mathcal{N}(0, I), & t > 1 \\ 0, & t = 1 \end{cases}$; |
| $\quad\quad \mathbf{x}^{\text{LR}} + \sqrt{\bar{\alpha}_t}(\mathbf{x}^{\text{HR}} - \mathbf{x}^{\text{LR}}) + \sqrt{1 - \bar{\alpha}_t}\boldsymbol{\epsilon}$; | $\quad \hat{\mathbf{x}}_0 \leftarrow f_\theta(\mathbf{x}_t, \text{concat}(\mathbf{x}^{\text{LR}}, \mathbf{m}), t)$; |
| $\quad$ // Concat noise & condition | $\quad \mathbf{r}_{t-1} \leftarrow$ |
| $\quad \hat{\mathbf{x}}_0 \leftarrow f_\theta(\mathbf{x}_t, \text{concat}(\mathbf{x}^{\text{LR}}, \mathbf{m}), t)$; | $\quad\quad \tilde{\mu}(\mathbf{x}_t - \mathbf{x}^{\text{LR}}, \hat{\mathbf{x}}_0 - \mathbf{x}^{\text{LR}}, t) + \sqrt{\tilde{\beta}_t}\boldsymbol{\epsilon}$; |
| $\quad \theta \leftarrow \theta - \eta \nabla_\theta \|\hat{\mathbf{x}}_0 - \mathbf{x}^{\text{HR}}\|^2$; | $\quad \mathbf{x}_{t-1} \leftarrow \mathbf{x}^{\text{LR}} + \mathbf{r}_{t-1}$; |
| **until** *converged*; | **end** |
| | **return** $\mathbf{x}_0$; |

## 3.2. Orientation-Conditioned FiLM Modulation

While deep networks may implicitly infer slice orientation from anatomical semantics, we explicitly condition the backbone on the viewing plane $y \in \{\text{axial, coronal, sagittal}\}$ as a direct inductive bias for domain adaptation. Slices acquired from different planes follow distinct 2D appearance distributions due to anisotropic sampling, orientation-dependent blur, and partial-volume effects. By injecting $y$ as a conditioning token, the shared 2D backbone dynamically adapts its feature distributions in a view-specific manner, without

introducing heavy computational overhead or parameter growth associated with parallel expert-style architectures such as MoE or ControlNet.

The categorical orientation label is embedded and mapped to affine modulation parameters $(\boldsymbol{\gamma}, \boldsymbol{\beta}) \in \mathbb{R}^D$ through a lightweight two-layer MLP, where $D$ denotes dimensionality of the diffusion timestep embedding. These parameters are applied to timestep embedding $\mathbf{e}_t$ via FiLM, yielding a view-conditioned embedding that is injected into each residual block. To control the strength of orientation conditioning across network depth, we introduce *depth-adaptive gating*, where a learnable scalar $g_i \in (0, 1)$ modulates the injection at the $i$-th residual block:

$$\hat{\mathbf{e}}_{t,i} = \mathbf{e}_t \odot \left(1 + \alpha \cdot g_i \cdot \boldsymbol{\gamma}\right) + \alpha \cdot g_i \cdot \boldsymbol{\beta}, \tag{2}$$

where $\alpha$ is a global annealing factor linearly increased during early training phase. The modulated embedding $\hat{\mathbf{e}}_{t,i}$ is then passed to the $i$-th residual block following standard diffusion U-Net formulation. This gating mechanism allows network to autonomously optimize conditioning strength, selectively activating view-specific corrections at beneficial depths while suppressing modulation where invariant representations are preferred.

### 3.3. Multi-level Data Augmentations

Given scarcity of paired fetal MRI data, we employ a comprehensive augmentation pipeline designed to enforce invariance to pose and robustness to acquisition artifacts. Following Zimmermann (2025), we model the degradation function $\mathcal{D}$ as a composition of geometric, intensity, and measurement perturbations applied on-the-fly to $(\mathbf{x}^{\mathrm{LR}}, \mathbf{x}^{\mathrm{HR}})$ pairs. Specific augmentation parameters are detailed in Appendix A.

**Geometric Invariance.** To simulate diverse fetal poses and preserve alignment, synchronized spatial transformations $\mathcal{T}_{geom}$ are applied to the input triplet $(\mathbf{x}^{\mathrm{LR}}, \mathbf{x}^{\mathrm{HR}}, \mathbf{m})$. These include random rotations, horizontal/vertical flips, and affine perturbations (shear and translation). Additionally, low-frequency B-spline deformations are applied to model non-rigid maternal/fetal motion.

**Intensity and Artifact Robustness.** We introduce realistic signal degradations to minimize the domain shift between synthetic training data and clinical inputs. These include: *Contrast Shifts* applying a non-linear monotonic intensity mapping via piecewise-linear interpolation through random control points to simulate scanner-specific contrast variations; *Signal Dropout* to emulate motion-induced signal voids common in single-shot HASTE, using "blackout" corruptions as a strong regularization surrogate: $\mathbf{x}_{\mathrm{corrupt}} = \mathbf{x} \odot (1 - \mathbf{M}_{\mathrm{b}}) + v_{\min} \cdot \mathbf{M}_{\mathrm{b}}$, where $\mathbf{M}_{\mathrm{b}}$ uniformly sample from four masking topologies (hemispheric, vertical band, oblique stripe, and multi-patch dropout); and *Measurement Noise* where variable Gaussian noise and anisotropic blurring are injected to approximate coil-dependent Rician noise and slice-thickness–induced point-spread effects.

### 3.4. Training Objective

Since our network $f_\theta$ directly predicts the clean estimate $\hat{\mathbf{x}}_0$, we use a combined reconstruction and perceptual loss. The primary objective is an $\ell_2$ loss on the prediction:

$$\mathcal{L}_{\text{diff}} = \mathbb{E}_{\mathbf{x}^{\text{HR}}, \mathbf{x}^{\text{LR}}, \mathbf{m}, \boldsymbol{\epsilon}, t, y} \left[ \left\| f_\theta(\mathbf{x}_t, \text{concat}(\mathbf{x}^{\text{LR}}, \mathbf{m}), t, y) - \mathbf{x}^{\text{HR}} \right\|_2^2 \right] \tag{3}$$

where the expectation is taken over the data distribution, diffusion noise $\boldsymbol{\epsilon}$, timesteps $t$, and orientation labels $y$. To ensure perceptual fidelity and textural sharpness, we add an LPIPS term, yielding the final objective, with $\lambda_{\text{mse}}$ and $\lambda_{\text{lpips}}$ balancing the two terms:

$$\mathcal{L}_{\text{total}} = \lambda_{\text{mse}} \mathcal{L}_{\text{diff}} + \lambda_{\text{lpips}} \text{LPIPS}(\hat{\mathbf{x}}_0, \mathbf{x}^{\text{HR}}), \tag{4}$$

## 4. Experiments

### 4.1. Datasets and Implementation

We utilized both synthetic and clinical $T_2$ HASTE datasets for model development and evaluation. Full training hyperparameters and details are provided in Appendix A.

**Synthetic Data.** Our primary training dataset comprises paired synthetic HR and LR fetal $T_2$ HASTE MRI volumes. These, alongside their corresponding tissue segmentation maps, were all generated using the spatiotemporal IMAGINE atlas (21–38 weeks gestational age (GA)) (Gholipour et al., 2023) and the FaBiAN numerical phantom (Lajous et al., 2022). FaBiAN employs a fast spin-echo (FSE) model to simulate extended phase-graph physics, stimulated echoes, and bias fields, accurately mimicking clinical HASTE acquisitions. For each GA and orientation, we synthesized clean 3T reference stacks ($B_0 = 3$T, $TE_{\text{eff}} = 90$ms, 0.3mm gap) and matched degraded 1.5T counterparts ($B_0 = 1.5$T, $TE_{\text{eff}} = 133$ms, 0mm gap) both with 3mm slice thickness. The 1.5T inputs were further augmented with variable motion, TE perturbations, and noise to reflect clinical heterogeneity and preserve underlying atlas anatomy. These simulated acquisition parameters align with our clinical fetal HASTE protocol, ensuring the 3T and 1.5T images replicate characteristic contrast and resolution. The dataset was split 80/20 for training and testing, with training slices undergoing the additional augmentation pipeline described in Section 3.3. We empirically validate the realism of these simulations by comparing their intensity distributions against the clinical target domain in Appendix E.

**Clinical Data.** For clinical validation, we utilized an IRB-approved clinical 3T $T_2$-weighted HASTE dataset from Children's Healthcare of Atlanta (CHOA), acquired with a voxel size of $0.98 \times 0.98 \times 3.0$mm$^3$. Following manual QC (206/244 subjects retained) and brain extraction (Ranzini et al., 2021), we designated 103 subjects with fully intact stacks as held-out test set (664 stacks) and remaining 103 subjects as validation pool (464 stacks) to monitor training. To simulate low-field physics, we applied spectral k-space truncation (Chen et al., 2018)—zero-filling high frequencies in Fourier domain—which approximates physical point-spread blurring more faithfully than image-domain downsampling.

## 4.2. Competing Methods

We benchmark against four state-of-the-art MRI super-resolution methods: **SRCNN** (Dong et al., 2015), **Real-ESRGAN** (Wang et al., 2018), **SwinIR** (Liang et al., 2021), and the 3D state-space model **GAMBAS** (Baljer et al., 2025). All baselines were retrained on our FaBiAN dataset using identical splits and optimized to convergence; detailed specifications are provided in Appendix B.

## 4.3. Evaluation Metrics

To evaluate the volumetric consistency of the 2D super-resolved stacks, output slices from all methods are first reconstructed into isotropic volumes using NeSVoR (Xu et al., 2023) with fixed hyperparameters. We then assess both reconstruction fidelity and downstream anatomical utility under synthetic and clinical settings.

On FaBiAN test set, we utilize the paired ground truth to compute standard restoration metrics: PSNR, NRMSE, and MAE measure intensity accuracy, while SSIM and LPIPS quantify structural and perceptual realism. To evaluate downstream utility, reconstructed volumes are segmented using Fetal-SynthSeg (Zalevskyi et al., 2024). We report region-based (Dice, Volume Bias) and boundary-based (ASSD, HD95) metrics against the simulation's ground truth tissue maps to assess anatomical integrity.

For CHOA-3T dataset, where pixel-aligned references are unavailable, we employ Tissue Contrast T-score (TCT) (Sun et al., 2025) to quantify separability of White Matter (WM) and Gray Matter (GM) distributions:

$$TCT = \frac{|\mu_{wm} - \mu_{gm}|}{\sqrt{\sigma_{wm}^2 + \sigma_{gm}^2}}, \tag{5}$$

where $\mu$ and $\sigma^2$ denote the mean and variance of tissue intensities derived from Fetal-SynthSeg masks. 1.5T acquisitions typically exhibit lower SNR, leading to increased intra-tissue variance ($\mu^2$) in the denominator. A higher TCT therefore indicates successful 3T-like super-resolution, driven by widened contrast separation ($|\mu_{wm} - \mu_{gm}|$) and suppressed noise. To eliminate confounders arising from biological maturation (e.g., myelination), we strictly limit TCT comparisons to within the same GA.

## 5. Results

Ablation studies validating individual components are detailed in Appendix C. An evaluation of inference efficiency and memory usage is provided in Appendix D.

## 5.1. Quantitative Validation on Synthetic Data

We first evaluate reconstruction fidelity and downstream utility on the FaBiAN test set.
**Reconstruction Fidelity.** As shown in Table 1, our method achieves the best perceptual metrics (**SSIM 0.81, LPIPS 0.042**). While SwinIR yields slightly higher PSNR due to its regression-based objective, it suffers from characteristic over-smoothing. In contrast, our model balances fidelity and realism, avoiding the artifacts seen in GAN baselines (Real-ESRGAN) and the degradation of GAMBAS under domain shifts. Appendix C.3

Table 1: Volumetric reconstruction metrics on FaBiAN synthetic test set.

| Method | PSNR (dB) ↑ | NRMSE ↓ | MAE ↓ | SSIM ↑ | LPIPS ↓ |
|---|---|---|---|---|---|
| Input (Sim. 1.5T) | 24.90 ± 2.10 | 0.36 ± 0.02 | 29.50 ± 5.90 | 0.79 ± 0.06 | 0.055 ± 0.038 |
| SRCNN | 25.15 ± 1.95 | 0.35 ± 0.02 | 28.80 ± 5.40 | 0.80 ± 0.05 | 0.052 ± 0.035 |
| Real-ESRGAN | 23.40 ± 2.45 | 0.42 ± 0.04 | 34.10 ± 7.20 | 0.72 ± 0.08 | 0.048 ± 0.041 |
| SwinIR | **26.12 ± 1.85** | **0.31 ± 0.02** | **26.50 ± 5.10** | 0.73 ± 0.05 | 0.047 ± 0.032 |
| GAMBAS | 14.80 ± 3.50 | 0.93 ± 0.03 | 176.20 ± 6.40 | 0.53 ± 0.07 | 0.158 ± 0.060 |
| **Ours** | 25.66 ± 2.06 | 0.34 ± 0.02 | 27.31 ± 5.70 | **0.81 ± 0.06** | **0.042 ± 0.037** |

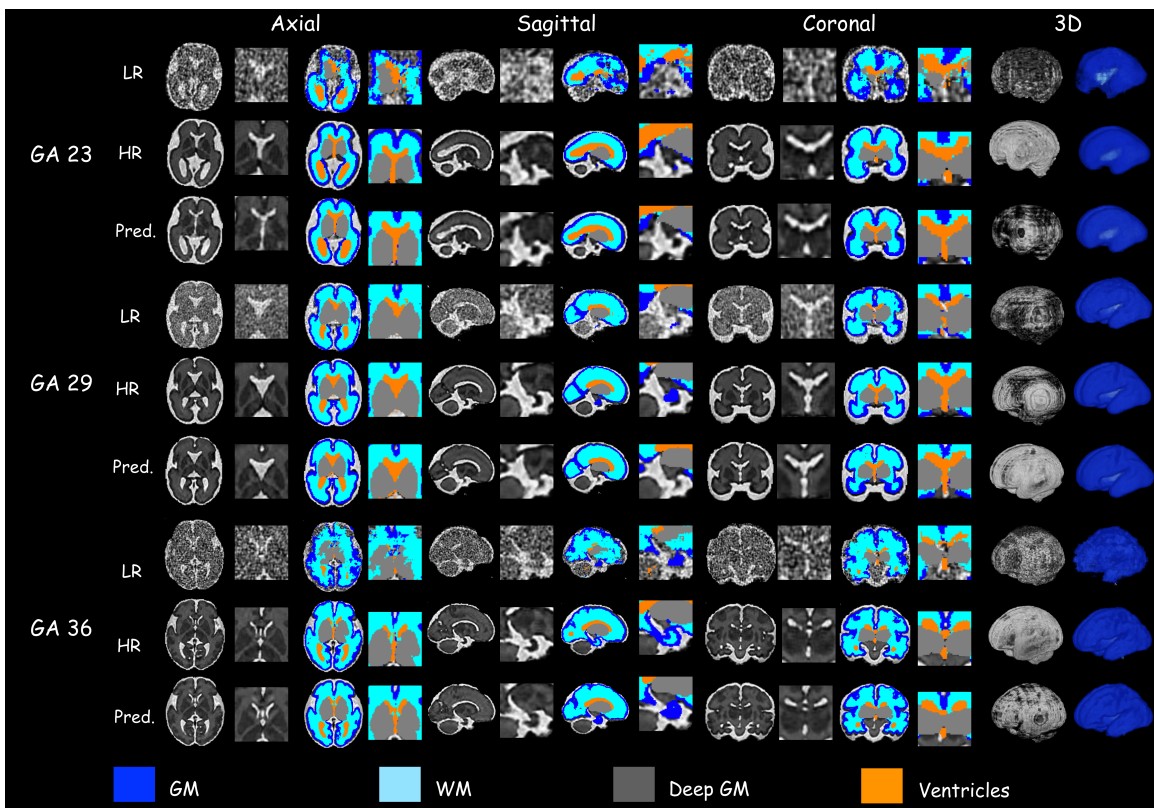

Figure 2: Qualitative evaluation on the held-out synthetic FaBiAN dataset across three representative gestational ages (23, 29, and 36 weeks).

confirms these architectural gains persist even when baselines are retrained with our full augmentation suite.

**Downstream Segmentation.** Improved image quality translates directly to segmentation accuracy (Table 2). Our method consistently outperforms baselines, with the largest gains observed in intricate structures such as *GM* (**Dice +28% relative to input**) and *Ventricles*. This corroborates the visual recovery of fine gyral patterns shown in Figure 2.

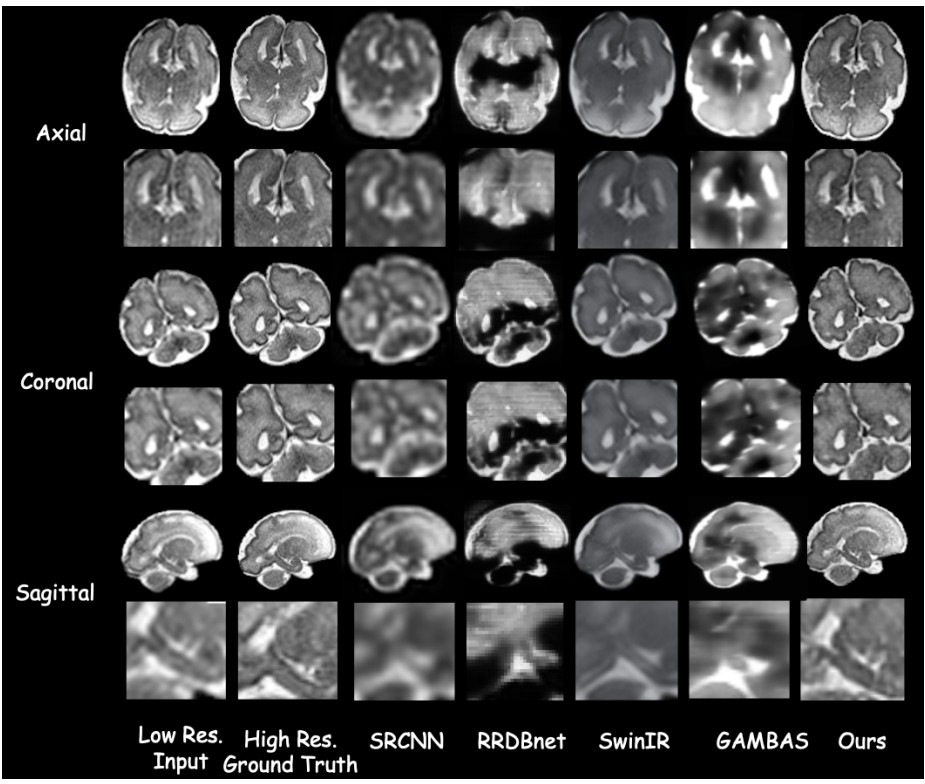

Figure 3: Comparative results on a clinical CHOA test subject.

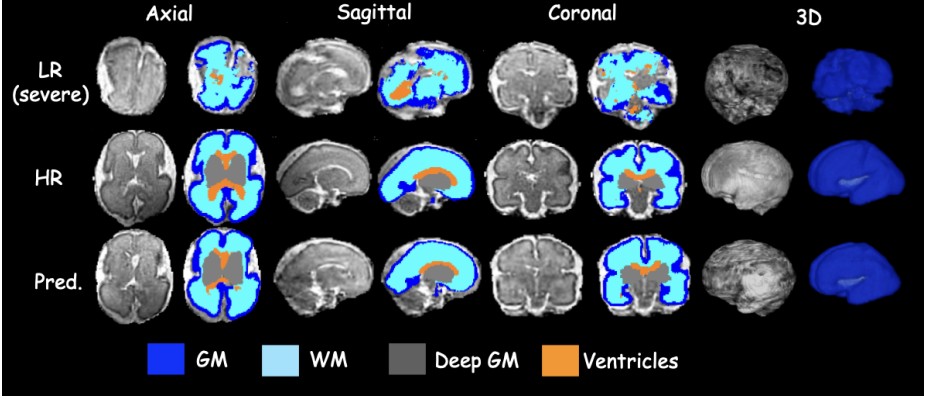

Figure 4: Restoration of a severely degraded case exhibiting motion and signal dropout.

Table 2: Region-wise segmentation metrics on FaBiAN synthetic test set.

| Region | Metric | Input (Sim. 1.5T) | SRCNN | RealESR-GAN | SwinIR | GAMBAS | Ours |
|---|---|---|---|---|---|---|---|
| GM | Dice ↑ | 0.551 ±0.030 | 0.586 ±0.028 | 0.495 ±0.045 | 0.658 ±0.032 | 0.352 ±0.051 | **0.704 ±0.024** |
| | ASSD ↓ | 0.521 ±0.034 | 0.486 ±0.065 | 0.853 ±0.123 | 0.396 ±0.029 | 1.452 ±0.180 | **0.334 ±0.021** |
| | HD95 ↓ | 1.414 ±0.069 | 1.351 ±0.157 | 2.124 ±0.210 | 1.057 ±0.085 | 3.214 ±0.420 | **0.853 ±0.075** |
| | Vol. Bias | -7.20 ±0.24 | -5.61 ±0.31 | +3.42 ±0.55 | -2.10 ±0.18 | -15.33 ±1.20 | **-0.55 ±0.14** |
| WM | Dice ↑ | 0.674 ±0.040 | 0.705 ±0.035 | 0.621 ±0.048 | 0.795 ±0.038 | 0.512 ±0.065 | **0.837 ±0.022** |
| | ASSD ↓ | 0.580 ±0.038 | 0.544 ±0.063 | 0.753 ±0.099 | 0.440 ±0.032 | 1.198 ±0.154 | **0.382 ±0.025** |
| | HD95 ↓ | 1.412 ±0.089 | 1.382 ±0.114 | 1.954 ±0.181 | 1.157 ±0.095 | 2.946 ±0.310 | **0.984 ±0.082** |
| | Vol. Bias | -3.70 ±0.12 | -3.13 ±0.18 | +4.53 ±0.35 | **-1.07 ±0.15** | -9.45 ±0.85 | -1.42 ±0.11 |
| Deep GM | Dice ↑ | 0.819 ±0.040 | 0.838 ±0.038 | 0.762 ±0.057 | 0.885 ±0.035 | 0.643 ±0.072 | **0.895 ±0.019** |
| | ASSD ↓ | 0.705 ±0.033 | 0.686 ±0.055 | 0.924 ±0.085 | **0.374 ±0.030** | 1.547 ±0.208 | 0.395 ±0.028 |
| | HD95 ↓ | 1.502 ±0.099 | 1.458 ±0.124 | 1.856 ±0.160 | 1.134 ±0.090 | 2.805 ±0.354 | **0.952 ±0.085** |
| | Vol. Bias | -3.75 ±0.15 | -2.59 ±0.21 | +6.85 ±0.40 | **+0.85 ±0.12** | +10.20 ±0.95 | +1.02 ±0.09 |
| Ventricles | Dice ↑ | 0.844 ±0.039 | 0.860 ±0.036 | 0.793 ±0.055 | 0.895 ±0.032 | 0.694 ±0.068 | **0.905 ±0.021** |
| | ASSD ↓ | 0.651 ±0.044 | 0.616 ±0.060 | 0.953 ±0.114 | **0.364 ±0.041** | 1.612 ±0.220 | 0.392 ±0.037 |
| | HD95 ↓ | 1.490 ±0.089 | 1.481 ±0.143 | 2.106 ±0.252 | 1.148 ±0.080 | 3.789 ±0.455 | **0.981 ±0.075** |
| | Vol. Bias | -7.11 ±0.33 | -5.24 ±0.28 | -8.53 ±0.45 | -2.09 ±0.22 | -14.39 ±1.10 | **-1.75 ±0.16** |

Table 3: TCT Scores on CHOA-3T clinical dataset.

| Method | Input (Sim. 1.5T) | SRCNN | RealESR-GAN | SwinIR | GAMBAS | Ours |
|---|---|---|---|---|---|---|
| **TCT** ↑ | $0.78 \pm 0.05$ | $0.82 \pm 0.04$ | $0.90 \pm 0.07$ | $0.88 \pm 0.03$ | $0.65 \pm 0.08$ | **$0.94 \pm 0.02$** |

### 5.2. Clinical Generalization on CHOA Dataset

We validate clinical performance by comparing reconstruction quality (Figure 3) and quantitative tissue contrast (Table 3). Qualitatively, our orientation-aware diffusion model preserves the contrast gradients essential for diagnosis, avoiding the smoothing artifacts of regression-based baselines. This is confirmed by the TCT metric, where our method achieves the highest score ($\mathbf{0.94 \pm 0.02}$), indicating superior gray-white matter separability. Finally, robustness to extreme corruption is demonstrated in Figure 4, where our method restores anatomical coherence from a heavily degraded input.

## 6. Discussion

Our work effectively bridges the domain gap between routine 1.5T acquisitions and diagnostic-quality 3T imaging. A critical innovation is the explicit modeling of acquisition anisotropy: unlike standard models that treat axial, sagittal, and coronal stacks identically—often resulting in isotropic blurring—our framework conditions the diffusion process on slice orientation to invert view-specific degradations. This restoration improves downstream utility, yielding robust 3D reconstructions and precise tissue segmentation. Quantitatively, our method achieves a TCT score of 0.94 on clinical CHOA dataset (vs. 0.78 for input), indicating a significant recovery of gray-white matter contrast comparable to high-field imaging.

However, we acknowledge certain limitations. Potential failure modes may arise if clinical inputs contain artifacts strictly outside our simulated training distribution. While our current validation on the CHOA dataset demonstrates strong generalization, future work will aim to curate a multi-institutional dataset spanning diverse hardware vendors (e.g., Siemens, GE, Philips) to further stress-test these boundaries and ensure reliability across different scanner manufacturers.

## Acknowledgments

This work was supported by NIH grants R00HD103912 and R01MH133313 (Y.W.).

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

## Appendix A. Implementation of Our Method

Our framework is implemented in PyTorch 2.5.1 and trained on three NVIDIA H100 GPUs. We utilize the FaBiAN-derived synthetic pairs with the augmentation suite detailed in Section 3.3. During training, we draw random $128 \times 128$ patches, while inference uses $256 \times 256$ center crops; all inputs are symmetrically padded and normalized to $[-1, 1]$. To simulate signal dropout, we apply blackout corruption with a probability of 0.5 (fill value -1.0) and enable monotonic intensity remapping with a probability of 0.2. Optimization is performed using RAdam with a batch size of 64 for 182k iterations. The learning rate follows a cosine-annealed schedule (base $2\times10^{-5}$ to $5\times10^{-5}$) after a 5k-step warm-up. We set $\lambda_{\mathrm{mse}} = 4.0$ and $\lambda_{\mathrm{lpips}} = 1.0$, and anneal the FiLM modulation strength over the first 50k iterations. At inference, we use the residual-shift sampler with a spatial scale factor $s_f = 1.0$, an exponential noise schedule (power 0.3), $\eta_{\mathrm{end}} = 0.99$, 4 diffusion steps, minimum noise level 0.2, and $\kappa = 2.0$.

### A.1. Data Augmentation Hyperparameters

We apply synchronized geometric transformations with the following parameters: random rotations sampled from $k \cdot 90°$ and $\mathcal{U}(-10°, 10°)$; shearing from $\mathcal{U}(-0.05, 0.05)$; and translations within $\pm 8$ pixels. Intensity remapping is applied with probability $p = 0.2$ using piecewise-linear functions through random control points. Signal dropout (blackout) is applied with probability $p = 0.5$ using a fill value of -1.0, masking regions with hemispheric, vertical band, oblique stripe, or multi-patch topologies.

## Appendix B. Implementation of competing methods

**SRCNN.** As a lightweight convolutional baseline we use a modified SRCNN-style architecture consisting of an initial $3 \times 3$ convolution followed by two DenseBlocks, each containing four dense layers with growth rate 32 and a transition layer reducing the feature dimensionality back to 64 channels. Two residual blocks refine the features before reconstruction. Upsampling is performed using bilinear interpolation with scale factor 4, followed by a final $3 \times 3$ convolution to produce the single–channel output. The network is trained with the pixel-wise MSE loss and optimized using Adam with learning rate $\mathrm{lr} = 1 \times 10^{-4}$.

**Real-ESRGAN.** We use a standard ESRGAN generator configured for scale 1 quality enhancement. The network contains 64 base feature channels and 23 residual-in-residual dense blocks (RRDBs), each employing residual dense connections with a scaling factor of 0.2. No discriminator or adversarial objective is used; instead, the model is trained purely as a feed-forward regressor with the pixel loss in Equation (4). Optimization uses Adam with an initial learning rate of $2 \times 10^{-4}$ and $\beta = (0.9, 0.99)$, together with a two-stage decay at 50k and 100k iterations. Training is run for 200k iterations with random $256 \times 256$ paired fetal MRI patches and standard flip/rotation augmentations.

**SwinIR.** We adopt the SwinIR "restoration" configuration (upscale = 1) for medical image quality enhancement. The model takes single–channel grayscale inputs and uses $256 \times 256$ patches with a window size of 8. The backbone follows the standard SwinIR-M design with six residual Swin Transformer blocks (RSTBs), each using six Swin layers with shifted windows. We set the embedding dimension to 180, use six attention heads per stage, an

Table 4: Ablation of residual-shift diffusion versus image-space diffusion on the FaBiAN test set.

| Variant | PSNR (dB) ↑ | NRMSE ↓ | MAE ↓ | SSIM ↑ | LPIPS ↓ |
|---|---|---|---|---|---|
| Vanilla DDPM (image-space) | 20.02 ±4.43 | 0.87 ±0.04 | 55.60 ±6.22 | 0.70 ±0.06 | 0.082 ±0.041 |
| Residual-shift | **25.66** **±2.06** | **0.34** **±0.02** | **27.31** **±5.70** | **0.81** **±0.06** | **0.042** **±0.037** |

MLP expansion ratio of 2, and employ "1conv" residual connections without any upsampling modules. Training uses Adam with a learning rate of $2 \times 10^{-4}$, batch size 32, and 500k iterations with a $\times 0.5$ decay every 150k.

**GAMBAS.** The Generalised-Hilbert Mamba SR model operates on short 3D volumes, so we input the stacks for each orientation and map them through eight Hilbert-ordered state-space layers with 96 hidden channels. Training uses Ranger (RAdam + Lookahead) with an initial learning rate $5 \times 10^{-4}$, cosine restarts every 50k steps, stochastic depth 0.1, and the same $\ell_1$+LPIPS objective as the planar baselines.

## Appendix C. Ablation Studies

We ablate the four main design choices of our framework on the synthetic FaBiAN test set and the CHOA clinical cohort.

### C.1. Residual vs. Image-space Diffusion.

We compare our residual-shift formulation to a vanilla image-space DDPM that directly diffuses the HR target while conditioning on the LR input via channel concatenation. Both variants share the same Swin-UNet backbone, noise schedule, and 4-step sampler. As shown in Table 4, residual diffusion maintains strong reconstruction performance, whereas the image-space DDPM degrades substantially across all metrics under this extreme low-step regime. This demonstrates that residual anchoring significantly improves sampling efficiency, enabling stable and accurate reconstruction with very few diffusion steps.

### C.2. Orientation Conditioning.

As shown in Table 5, removing orientation information leads to a clear degradation in Dice across all tissue classes. Additive embeddings provide limited gains, while FiLM modulation substantially recovers segmentation accuracy. The proposed depth-adaptive gated FiLM achieves the best overall performance, demonstrating the importance of selectively activating orientation-specific priors.

### C.3. Fetal-specific Augmentations.

We treat the proposed multi-level augmentation suite as a core contribution designed to bridge the synthetic-to-clinical domain gap. To verify that our performance gains stem from the synergy between our model architecture and this data strategy—rather than the

Table 5: Ablation of orientation conditioning on the FaBiAN test set. Mean denotes the average Dice across all tissue classes.

| Variant | GM | WM | Deep GM | Ventricles | Mean |
|---|---|---|---|---|---|
| w/o orientation | 0.602 ±0.031 | 0.742 ±0.036 | 0.851 ±0.028 | 0.866 ±0.030 | 0.765 |
| Additive embedding | 0.645 ±0.029 | 0.781 ±0.032 | 0.872 ±0.024 | 0.885 ±0.027 | 0.796 |
| FiLM (w/o gating) | 0.681 ±0.026 | 0.812 ±0.028 | 0.887 ±0.021 | 0.898 ±0.024 | 0.820 |
| Gated FiLM | **0.704** ±**0.024** | **0.837** ±**0.022** | **0.895** ±**0.019** | **0.905** ±**0.021** | **0.835** |

Table 6: Impact of Multi-level Augmentation on Baselines on the FaBiAN Test Set.

| Model | Augment. | PSNR(dB) ↑ | NRMSE ↓ | MAE ↓ | SSIM ↑ | LPIPS ↓ |
|---|---|---|---|---|---|---|
| SRCNN | Standard | 25.15 ±1.95 | 0.35 ±0.02 | 28.80 ±5.40 | 0.80 ±0.05 | 0.052 ±0.035 |
| | Multi-level | 25.61 ±1.73 | 0.33 ±0.03 | 28.80 ±5.90 | 0.76 ±0.06 | 0.049 ±0.037 |
| Real-ESRGAN | Standard | 23.40 ±2.45 | 0.42 ±0.04 | 34.10 ±7.20 | 0.72 ±0.08 | 0.048 ±0.041 |
| | Multi-level | 20.19 ±3.66 | 0.43 ±0.03 | 33.20 ±7.00 | 0.77 ±0.03 | 0.045 ±0.042 |
| SwinIR | Standard | 26.12 ±1.85 | **0.31** ±**0.02** | **26.50** ±**5.10** | 0.73 ±0.05 | 0.047 ±0.032 |
| | Multi-level | **26.21** ±**3.04** | 0.32 ±0.03 | 26.60 ±5.60 | 0.72 ±0.06 | 0.050 ±0.038 |
| GAMBAS | Standard | 14.80 ±3.50 | 0.93 ±0.03 | 176.20 ±6.40 | 0.53 ±0.07 | 0.158 ±0.060 |
| | Multi-level | 14.44 ±4.21 | 0.86 ±0.07 | 125.00 ±5.30 | 0.49 ±0.08 | 0.099 ±0.060 |
| **Ours** | Multi-level | 25.66 ±2.06 | 0.34 ±0.02 | 27.31 ±5.70 | **0.81** ±**0.06** | **0.042** ±**0.037** |

augmentations alone—we retrained all baselines using the full multi-level augmentation pipeline. As shown in Table 6, applying fetal-specific augmentations to baselines does not guarantee gains and can induce instability (e.g., Real-ESRGAN). While SwinIR maintains high pixel-wise accuracy, our method achieves superior structural preservation (highest SSIM) and perceptual realism (lowest LPIPS), confirming that our architecture is uniquely capable of leveraging these priors. Furthermore, Table 7 demonstrates that this strategy is indispensable for clinical deployment: disabling the augmentation suite significantly

Table 7: Impact of multi-level data augmentations on TCT on the CHOA clinical dataset.

| Variant | w/o Full Augment | Full Augment |
|---|---|---|
| **TCT** ↑ | $0.81 \pm 0.03$ | $\mathbf{0.94 \pm 0.02}$ |

degrades the TCT score on the CHOA dataset ($0.94 \rightarrow 0.81$), validating their role in facilitating robust sim-to-real transfer.

### C.4. LPIPS Loss.

To evaluate perceptual loss, we trained a variant with $\lambda_{\mathrm{LPIPS}} = 0$. Results in Table 8 confirm the perception-distortion trade-off (Blau and Michaeli, 2018): removing LPIPS marginally improves pixel-wise error (PSNR) but significantly degrades structural integrity (SSIM 0.81 vs. 0.77) and perceptual realism (LPIPS 0.042 vs. 0.063). Given that learning-based metrics align better with expert radiological assessment (Khateri et al., 2025), we retain the perceptual loss to prioritize the structural fidelity and sharpness essential for 3T synthesis.

Table 8: Impact of LPIPS perceptual Loss on FaBiAN Test Set.

| Variant | PSNR (dB) ↑ | NRMSE ↓ | MAE ↓ | SSIM ↑ | LPIPS ↓ |
|---|---|---|---|---|---|
| w/o LPIPS | **26.19** $\pm\mathbf{1.92}$ | 0.34 $\pm 0.05$ | **25.78** $\pm\mathbf{4.65}$ | 0.77 $\pm 0.06$ | 0.063 $\pm 0.050$ |
| **Ours** | 25.66 $\pm 2.06$ | **0.34** $\pm\mathbf{0.02}$ | 27.31 $\pm 5.70$ | **0.81** $\pm\mathbf{0.06}$ | **0.042** $\pm\mathbf{0.037}$ |

## Appendix D. Computational Efficiency

Table 9: Inference efficiency on a single NVIDIA A100 (batch size=1). Our method matches GAN-level latency while outperforming baselines in reconstruction quality.

| Model | Params (M) | Time (s) | Memory (GB) |
|---|---|---|---|
| SRCNN | 0.43 | 0.001 | 0.18 |
| Real-ESRGAN | 16.70 | 0.065 | 1.18 |
| SwinIR | 11.50 | 0.027 | 0.31 |
| GAMBAS | 53.45 | 0.046 | 3.31 |
| Vanilla DDPM | 56.67 | 18.054 | 0.92 |
| **Ours** | 56.67 | 0.079 | 0.92 |

To validate clinical feasibility, we benchmarked inference performance on a single NVIDIA A100 GPU (batch size=1). Despite being diffusion-based, our Residual-Shift formulation converges in just 4 steps, significantly reducing latency compared to standard DDPMs ($\sim$1,000 steps) or LDMs (50+ steps).

As shown in Table 9, our method requires only 0.079 seconds per slice, achieving speeds comparable to single-pass GANs (e.g., Real-ESRGAN: 0.065s) and $\approx 230\times$ faster than vanilla DDPMs. Furthermore, our efficient Swin-UNet backbone maintains a minimal memory footprint (0.92 GB)—lower than even the Real-ESRGAN baseline (1.18 GB)—due to window-based self-attention. This confirms that our approach is lightweight enough for real-time deployment on standard clinical workstations.

## Appendix E. Validation of Simulation Assumptions

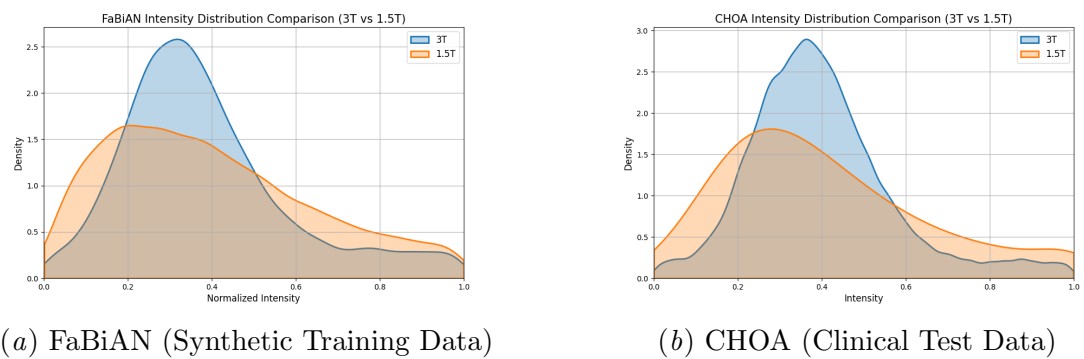

(*a*) FaBiAN (Synthetic Training Data)  (*b*) CHOA (Clinical Test Data)

Figure 5: **Intensity Distribution Analysis.** Comparison of voxel intensity densities between synthetic and clinical domains.

To validate our simulation, we compared the intensity distributions of synthetic (FaBiAN) training data against clinical (CHOA) test data. As shown in Figure 5, the synthetic profiles (a) closely align with the clinical data (b), confirming the representativeness of our training set. Furthermore, both domains consistently demonstrate the expected physical contrast gain: 3T targets (Blue) exhibit significantly sharper, higher-density signal peaks compared to the broader distributions of 1.5T inputs (Orange), validating the modeled relationship between field strengths.

