# OpenReview forum: "Orientation-Aware Diffusion Super-Resolution for 3T-Like Fetal MRI from Routine 1.5T Scans"
_MIDL.io/2026/Conference — MIDL 2026 Poster_

### Official Review · Reviewer_xBYa · 2025-12-23

**Confidence:** 4
**Preliminary Rating:** 4
**Final Rating:** 4

**Summary:**

The paper presents a method for fetal MRI super-resolution using diffusion models, where the generation process is conditioned on the orientation of the slices. This allows the model to account for different degradations occurring across various acquisition planes. The orientation information is incorporated using a feature modulation approach (FiLM), and the method also leverages extensive data augmentation to enhance performance. Experimental results demonstrate that the proposed approach performs well on both simulated and clinical datasets.

**Strengths:**

There are several strengths of the paper are:

1. The paper is well-motivated, offering a clear and concise description of both the application and the proposed methodology. The rationale for training diffusion models conditioned on different orientations is logical and convincing, with ablation results and quantitative evaluations in Table 5 supporting this assumption.

2.The evaluation on synthetic data using PSNR, MAE, SSIM, and LPIPS, along with segmentation metrics for the downstream task, is thorough and convincing.

3. The paper is well-written, easy to follow, and the inclusion of results on clinical data adds value.

**Weaknesses:**

Some Weaknesses in my opinion :

1. The novelty of the proposed method primarily lies in conditioning the generation process on orientation, which I believe is somewhat limited. However, given that the results are strong, I consider this a minor weakness.

2. It is unclear whether the comparison models were trained with the same data augmentation. From the implementation details, it does not appear so. If this is the case, is the comparison truly fair? For example, if SWIN-IR were trained with similar data augmentation, would it perform comparably or worse than the proposed method?

3. Ablation studies without the LPIPS loss are missing. While LPIPS can improve perceptual quality, it may reduce PSNR and MAE. Including an ablation study that controls the contribution of LPIPS could provide more balanced quantitative and qualitative results. Moreover, in MRI, smaller LPIPS values may not be as critical since MR images have limited texture variation compared to natural images. Setting LPIPS loss to zero might still yield reasonably good results.

4. A brief explanation or justification of why a higher TCT-score indicates better performance would be helpful.

5. Another way to evaluate performance on clinical data could be to intentionally downsample high-resolution images and test whether the model can recover the missing information. The authors argue that high-field 3T images are unavailable, but downsampling the existing 1.5T images further could serve as a valid test for the model’s reconstruction ability. Why is this not tested ?

**Detailed Comments:**

No major comments.

**Justification Of Final Rating:**

The paper is well aligned with the MIDL audience, and the experimental evaluation is convincing, with adequate comparisons to build confidence in the proposed approach. While I selected Weak Accept due to the available scoring options, the paper would warrant a slightly higher score than Weak Accept if such an option existed.

**Justification Of The Preliminary Rating:**

Although the methodology is not particularly novel, I believe the MIDL audience would appreciate the content of the paper. The application is relevant, and the results are strong, supported by convincing ablation studies. Therefore, I recommend it for a poster presentation.

**Questions To Address In The Rebuttal:**

Addressing weaknesses 2–5 should be a high priority in the rebuttal. I would ask the authors to clarify whether the same data augmentation was applied to the comparison models, and if not, provide a justification for this choice. If feasible within the available time, an ablation study evaluating the impact of LPIPS loss should be included. Additionally, the authors should justify their decision not to perform intentional downsampling experiments, since this approach is commonly used in many super-resolution deep learning studies.

---

> ### Author Response · Authors · 2026-01-24
>
> We thank the reviewer for endorsing our motivation and recognizing the work's suitability for MIDL. We have updated the manuscript to address the constructive suggestions regarding experimental design and metric clarification.
>
> **Fairness of Data Augmentation:** We agree that isolating architectural gains is critical. To address this, we retrained all baselines using our full multi-level augmentation suite (geometric, intensity, and motion corruption) on the FaBiAN test set (Appendix C.3 Table 6):
>
> | Model | Augmentation | PSNR(dB) $\uparrow$ | NRMSE $\downarrow$ | MAE $\downarrow$ | SSIM $\uparrow$ | LPIPS $\downarrow$ |
> | ----------- | ----------- | ----------- | ----------- | ----------- | ----------- | ----------- |
> | SRCNN | Standard  | 25.15 $\pm$ 1.95 | 0.35 $\pm$ 0.02 | 28.80 $\pm$ 5.40 | 0.80 $\pm$ 0.05 |0.052 $\pm$ 0.035 |
> |  | Multi-level | 25.61 $\pm$ 1.73 | 0.33 $\pm$ 0.03 | 28.80 $\pm$ 5.90 | 0.76 $\pm$ 0.06 | 0.049 $\pm$ 0.037 |
> | RealESR-GAN | Standard | 23.40 $\pm$ 2.45 | 0.42 $\pm$ 0.04 | 34.10 $\pm$ 7.20 | 0.72 $\pm$ 0.08 |0.048 $\pm$ 0.041 |
> |  | Multi-level | 20.19 $\pm$ 3.66 | 0.43 $\pm$ 0.03 | 33.20 $\pm$ 7.00 | 0.77 $\pm$ 0.03 | 0.045 $\pm$ 0.042 |
> | SwinIR | Standard  | 26.12 $\pm$ 1.85 | **0.31 $\pm$ 0.02** | **26.50 $\pm$ 5.10** | 0.73 $\pm$ 0.05 | 0.047 $\pm$ 0.032 |
> |  | Multi-level | **26.21 $\pm$ 3.04** | 0.32 $\pm$ 0.03 | 26.60 $\pm$ 5.60 | 0.72 $\pm$ 0.06 | 0.050 $\pm$ 0.038 |
> | GAMBAS | Standard  | 14.80 $\pm$ 3.50 | 0.93 $\pm$ 0.03 | 176.20 $\pm$ 6.40 | 0.53 $\pm$ 0.07 | 0.158 $\pm$ 0.060 |
> |  | Multi-level | 14.44 $\pm$ 4.21 | 0.86 $\pm$ 0.07 | 125.00 $\pm$ 5.30 | 0.49 $\pm$ 0.08 | 0.099 $\pm$ 0.060 |
> | Ours | Multi-level | 25.66 $\pm$ 2.06 | 0.34 $\pm$ 0.02 | 27.31 $\pm$ 5.70 | **0.81 $\pm$ 0.06** | **0.042 $\pm$ 0.037** |
>
> Applying augmentation to existing baselines does not guarantee gains and can induce instability (e.g., RealESR-GAN). While SwinIR maintains high pixel-wise accuracy, our method achieves superior structural preservation (highest SSIM) and perceptual realism (lowest LPIPS), confirming that performance stems from the synergy between architecture and data strategy. We would also like to clarify that we treated the "Multi-level Augmentation Suite" (Section 3.3 and Appendix C.3) as one of the core contributions of this work, designed specifically to bridge the synthetic-to-clinical domain gap in fetal MRI. Therefore, in the original manuscript, we compared our full framework (Model + Augmentation) against existing methods using their standard training protocols (typical flip/rotation) to demonstrate the total system improvement over current SOTA practices.
>
> **LPIPS Ablation:** We appreciate this suggestion and have added a full ablation study ($λ_{LPIPS}=0$), incorporated into the revised manuscript (Appendix C.4) of the revised manuscript. The results confirm that LPIPS is essential for structural fidelity (SSIM 0.81 vs 0.77), validating our choice to prioritize perceptual realism over marginal pixel-wise gains.
>
> **TCT-Score:** We thank the reviewer for this query and have added the following justification to Section 4.3. The Tissue Contrast T-score (TCT) measures tissue differentiability normalized by noise: $TCT = \frac{|\mu_{wm} - \mu_{gm}|}{\sqrt{\sigma^2_{wm} + \sigma^2_{gm}}}$. In 1.5T scans, low SNR inflates intra-tissue variance $(\sigma^2)$ depressing the score. Conversely, effective super-resolution widens the contrast gap ($|\mu_{wm} - \mu_{gm}|$), increasing the numerator. We also want to clarify that TCT comparisons are only valid within the same GA, as biological maturation (e.g., myelination, water content changes) inherently alters tissue contrast independent of image quality $^{[1]}$.
>
> **Downsampling Experiments:** We appreciate this suggestion and clarify that we indeed validated recovery using the clinical 3T CHOA dataset (Sections 4.1, 4.3, 5.2). Instead of downsampling 1.5T images (which lack ground truth), we applied spectral k-space truncation to real 3T scans following ${[2]}$ to accurately mimic low-resolution acquisition physics. This establishes a "Simulated 1.5T $\to$ Real 3T" paired benchmark, allowing us to evaluate high-frequency recovery against a valid ground truth. While we noted that paired 3T references are unavailable for the routine 1.5T clinical workflow, this CHOA dataset was explicitly curated to serve as the valid test bed requested by the reviewer.
>
> [1] Prayer, Daniela, et al. "MRI of normal fetal brain development." European journal of radiology 57.2 (2006): 199-216.
>
> [2] Chen, Yuhua, et al. "Brain MRI super resolution using 3D deep densely connected neural networks." 2018 IEEE 15th international symposium on biomedical imaging (ISBI 2018). IEEE, 2018.

---

> ### Comment · Reviewer_xBYa · 2026-01-24
> **Reply to Authors comments**
>
> I find that the authors’ responses adequately address all the issues I raised. I encourage the authors to revise the paper to incorporate the additional details and results discussed above, either in the main manuscript or in the appendix, so that future readers do not encounter the same questions. I will update my rating accordingly based on this discussion. All the best.

---

> > ### Author Response · Authors · 2026-01-24
> >
> > Dear Reviewer xBYa,
> >
> > Thank you for your prompt response and encouraging feedback.
> >
> > We have included the details you requested in the **revised manuscript**, which is available at the top of this page in the **"Rebuttal by Authors"** section under **"Supporting Material"**.
> >
> > All changes within the document are **highlighted in red** for your convenience.
> >
> > Best regards,
> > The Authors

---

### Official Review · Reviewer_U53J · 2026-01-08

**Confidence:** 4
**Preliminary Rating:** 2
**Final Rating:** 2

**Summary:**

The paper proposes an orientation-aware diffusion super-resolution framework that synthesizes 3T-like fetal brain contrast from routine 1.5T scans, essentially translating 1.5T scans to 3T quality in terms of effective resolution.
The method addresses high-frequency limitations on image restoration by adopting a residual-shift formulation; the model iteratively refines input details, rather than synthesizing images from scratch.

**Strengths:**

The paper provides a clear and well-structured description of the problem. The use of FiLM-based layers to handle scan orientation is an interesting and creative design choice, even though orientation could in principle be resolved from header metadata. Also, a better description of their utility, for example, it may improve different slice thicknesses in different axes, could be described better.

**Weaknesses:**

The conceptual framework and training procedure are not clearly explained. The paper does not adequately define what X_LR, Xt_LR, Xt_HR, and masks represent or how they are obtained, both in Figure 1 and Section 3.1. Additionally, it is unclear what the Swin-UNet backbone refers to and how the feature extractor operates in Figure 1.
These problems make the paper difficult to understand and implement and of limited practical feasibility, which weakens its potential impact despite the originality of the underlying idea.

**Detailed Comments:**

More details:
•	Figure 1: Please increase the overall shape to fit the width of the text.
•	FiLM-based layers: why apply them to all the ResBlocks? A retrospective analysis by the original paper authors suggests applying early in the CNN, rather than all, to avoid overfitting.
•	It’s not clear what the role of the “Swin-UNet backbone” is. Is it the feature extractor in Figure 1? How was it trained?
•	Section 3.1 lacks critical information and is very difficult to understand. Not sure what the objective is here.
•	Figure 2: Please additionally display small brain sections instead of the full brain, as it is not possible to see the quality.

**Justification Of Final Rating:**

Despite the rebuttal, the core clarity issues remain largely unaddressed. Figure 1 is still confusing and shows only minimal changes, leaving the overall framework difficult to interpret. The methodological description did not improve substantially, and the role and analysis of the FiLM-gated layers remain unclear. As a result, it is still challenging to fully assess the validity and impact of the proposed approach.

**Justification Of The Preliminary Rating:**

The conceptual framework and training procedure are not clearly explained, which makes it difficult to assess the technical soundness and novelty of the method. Key components are insufficiently defined, and the data flow throughout the pipeline remains unclear. Without a clear understanding of how the model is constructed, trained, and evaluated, it is difficult to judge the rigor or impact of the work.

**Questions To Address In The Rebuttal:**

I would focus on explaining the main architecture and the training procedure in the rebuttal.

---

> ### Author Response · Authors · 2026-01-24
>
> We thank the reviewer for their thoughtful and detailed feedback. We apologize that the initial description of the framework caused confusion. We have revised the manuscript to address these issues.
>
>
> **Clarification of Framework and Variables:** We agree with the reviewer for correctly pointing out the need to better define the inputs and the framework. We have revised Section 3.1 and Figure 1 to define all variables and standardize the notation to avoid ambiguity:
>
> - $x_{LR}$: The fixed low-resolution input slice as a **condition**.
> - $m$: The binary validity mask, used as a **condition** to ensure the model focuses only on non-background brain regions.
> - $x_{t}$: The noisy **latent** variable at timestep $t$. In our Residual-Shift formulation, this state transitions from a noisy initialization based on the input $x_T \approx x_{LR} + \epsilon$ at the maximum timestep $T$ to the clean high-resolution estimate $x_{0}$. (Note: We have removed the confusing superscript "$LR$" from the noisy state $x_{t}$ in the revision).
> - $x_{HR}$: The clean ground truth **target** used for loss calculation.
>
> Regarding the reviewer's query on the "objective" in Section 3.1, we clarify that this refers to the diffusion parameterization. Unlike standard DDPMs that predict the noise term $\epsilon$, our Swin-UNet backbone is parameterized to predict the clean high-resolution estimate $\hat{x}$ directly at each timestep. The estimate approximates to the target $x_{HR}$, effectively allowing the network to recover the residual mapping $r_0 = \hat{x}_{0} - x^{LR}$ on top of the fixed input anchor.
>
> **Role of Swin-UNet Backbone:** To address the confusion regarding the "Swin-UNet backbone" versus the "feature extractor," we clarify that our framework utilizes a dual-stream input strategy typical of conditional diffusion models $^{[1]}$. Specifically, the **Conditioning Encoder** (labeled "Feature Extractor" in the original Figure 1) is a lightweight, shallow CNN that encodes the fixed reference inputs—the low-resolution slice $x_{LR}$ and validity mask $m$ — into spatial guidance features. In contrast, the **Denoising Backbone** $f_\theta$ is the deep Swin-UNet that accepts the concatenation of the noisy latent state $x_t$​ and these encoded features to perform the actual generative task. In the updated Figure 1, we have renamed "Feature Extractor" to "Conditioning Encoder" to visually distinguish it from the main backbone.
>
> **FiLM Layers on All ResBlocks:** We thank the reviewer for this insightful observation regarding the optimal placement of FiLM layers. We are aware of the retrospective analysis suggesting that feature-wise modulation is often most effective when restricted to early layers to prevent overfitting. Instead of manually hard-coding the placement (which risks being suboptimal), we introduced the Depth-Adaptive Gating mechanism (Eq. 2). By multiplying the FiLM parameters with a learnable scalar $g_i \in (0,1)$ initialized to zero, we allow the network to automatically learn the optimal topology. Our ablation study in Table 5 supports this design. The "Gated FiLM" model (Dice 0.835) outperformed the standard "FiLM w/o gating" (Dice 0.820). This performance gap confirms that the learned, soft-selection of layers provides a better balance between adaptation and regularization than simply applying it everywhere or restricting it manually.
>
> **Figure Formatting:** We thank the reviewer for these constructive suggestions to improve presentation clarity. In the revised manuscript, we have resized Figure 1 to span the full text width, ensuring the details are clearly legible. Additionally, we updated Figure 2 to include magnified ROIs alongside the full-brain slices.
>
> [1] Saharia, Chitwan, et al. "Image super-resolution via iterative refinement." IEEE transactions on pattern analysis and machine intelligence 45.4 (2022): 4713-4726.

---

### Official Review · Reviewer_2Pay · 2026-01-10

**Confidence:** 3
**Preliminary Rating:** 4

**Summary:**

A challenge in fetal  MRI is that 3T scanners with  higher SNR have lower motion tolerance, and thus clinical examinations are performed at 1.5T. The authors propose a method based on Swin-UNet to synthesize 3T-like fetal brain contrast from 1.5T scans. The proposed method improves tissue segmentation accuracy.

**Strengths:**

The empirical results are comprehensive, including both synthetic and real clinical data, multiple baselines, and well-designed ablations. The methodology is tailored to fetal MRI, using residual shift diffusion to mitigate hallucinations, explicit orientation conditioning via gated FiLM, and augmentations.

**Weaknesses:**

The authors do not compare FiLM-based orientation conditioning against simpler or hybrid alternatives in experiments. The paper lacks deeper analysis of failure modes or cases where simulation assumptions (simulated 1.5T–3T differences to capture real-world scanner variability) may break down in diverse clinical settings.

**Detailed Comments:**

Since the authors do not compare FiLM-based orientation conditioning against simpler or hybrid alternatives in experiments, it is hard to isolate whether the gains come specifically from gated FiLM versus orientation awareness in general.

**Justification Of The Preliminary Rating:**

The proosed orientation-aware residual diffusion for fetal MRI is novel. Although there are is a lack of practical deployment considerations (computational cost), this work is strong and well motivated.

**Questions To Address In The Rebuttal:**

The computational cost is not discussed. Could the authors discuss the complexity of the proposed method? What is the inference time cost and memory footprint?

---

> ### Author Response · Authors · 2026-01-24
>
> We thank the reviewer for their insightful comments on the paper, and for their positive feedback on the strengths of the work. We first address their questions and then their detailed suggested improvements to the paper and revise our manuscript accordingly.
>
> **Comparison of FiLM vs. Simple Alternatives:** We apologize if the placement of this analysis in the Appendix caused it to be overlooked. We would like to clarify that we indeed performed an ablation study comparing our method against simpler alternatives, such as Additive Embedding, in Table 5 (Appendix C.2) of the original submission.
> The results confirm that Gated FiLM significantly outperforms the simpler additive baseline (Mean Dice: 0.835 vs. 0.796), validating the necessity of affine modulation for handling complex geometric orientations. We are happy to include comparisons against other specific alternatives if requested by the reviewer.
>
>
> **Failure Modes:**
> We appreciate the reviewer's critique regarding the limits of our simulation assumptions and the need for a deeper analysis of potential failure modes. To quantitatively address whether our simulation captures real-world variability, we have added Appendix E (Figure 5) to the revised manuscript, which contrasts the intensity distributions of our synthetic training data (FaBiAN) against the clinical validation set (CHOA). As illustrated in the analysis, the synthetic and clinical distributions exhibit a remarkable morphological alignment, with both domains displaying the expected physical contrast gain (sharper signal peaks) when moving from 1.5T to 3T. While this broad coverage mitigates standard domain shifts, we have updated the Discussion to explicitly acknowledge that failure modes may still arise when clinical scans exhibit artifacts strictly outside our simulated distribution. To further stress-test these boundaries, we have outlined plans for future multi-vendor validation across Siemens, GE, and Philips platforms.
>
> **Computational Cost:**
> We thank the reviewer for highlighting deployment feasibility. To address this, we added a benchmarking analysis (Appendix D) on a single NVIDIA A100 GPU. Although diffusion-based, our Residual-Shift formulation converges in just 4 steps, reducing latency by $230\times$ compared to standard DDPMs. As shown below, our method processes a slice in 0.079s, effectively matching the speed of single-pass GANs (Real-ESRGAN: 0.065s). Furthermore, our Swin-UNet backbone is highly efficient, requiring only 0.92 GB of VRAM—surprisingly lower than the Real-ESRGAN baseline (1.18 GB) due to efficient window-based attention. This confirms our method bridges the gap between diffusion-level fidelity and the efficiency required for standard clinical workstations.
>
> | Model | Params (M) | Time per Slice (s) | Memory (GB) |
> | ----------- | ----------- | ----------- | ----------- |
> | SRCNN | 0.432 | 0.0010 | 0.175 |
> | RealESR-GAN | 16.70 | 0.0651 | 1.177 |
> | SwinIR | 11.50 | 0.0270 | 0.310 |
> | GAMBAS | 53.45 | 0.0461 | 3.309 |
> | Vanilla DDPM | 56.67 | 18.054 | 0.921 |
> | Ours | 56.67 | 0.0787 | 0.919 |

---

### Author Rebuttal · Authors · 2026-01-24

**Rebuttal:**

We sincerely thank the reviewers for their constructive feedback and encouragement. In the revised manuscript, we have clarified our methodology and evaluation metrics (Sections 3–4, Figures 1–2), incorporated an LPIPS ablation study (Appendix C.4), and provided computational efficiency benchmarking (Appendix D). Furthermore, to ensure a rigorous comparison, we retrained all baselines using our full augmentation suite (Appendix C.3 Table 6) and empirically validated our simulation assumptions against clinical data (Appendix E). Detailed point-by-point responses follow below.

Please find the revised manuscript (11 pages, with changes highlighted in red) included as Supporting Material.

**Supporting Material:**

/attachment/9c8039bbdc57e5242cee04f1665f19edbd18cd7e.pdf

---

### Meta-Review · Area_Chair_dBwT · 2026-02-07

**Recommendation:** Accept (Poster)
**Confidence:** 2

**Metareview:**

Two reviewers find the work solid and suitable for MIDL, highlighting strong empirical validation on both synthetic and clinical data, thoughtful architectural choices, and careful ablations. The orientation conditioning, residual-shift diffusion formulation, and downstream evaluation via fetal brain segmentation provide convincing evidence of practical value. The rebuttal was responsive and added important clarifications, additional ablations, and computational cost analysis, which substantially strengthened the paper.

One reviewer remains unconvinced, primarily due to clarity and presentation issues around the methodological description and architectural flow. While these concerns are valid and point to remaining weaknesses in exposition, the authors did make a good-faith effort to clarify notation, figures, and training procedures in the revision. From a meta perspective, the core technical contribution and experimental support appear sound, even if the presentation is not yet at an ideal level of clarity.

The paper sits near the acceptance threshold. The contribution is relevant to the MIDL audience, the results are OK, and the concerns raised by the more critical reviewer are largely about clarity rather than fundamental correctness. I therefore recommend accept, but borderline, with a poster presentation, and encourage the authors to further improve clarity and self-contained explanations in the final camera-ready version.

---

### Decision · Program_Chairs · 2026-02-13

Accept (Poster)